# The Application of Nanofibrous Resonant Membranes for Room Acoustics

**DOI:** 10.3390/nano13061115

**Published:** 2023-03-21

**Authors:** Klara Kalinova

**Affiliations:** Institute for Nanomaterials, Advanced Technologies and Innovation, Technical University of Liberec, 461 17 Liberec, Czech Republic; klara.kalinova@tul.cz; Tel.: +420-731-563-570

**Keywords:** nanofibers, sound absorption, grid, resonator, membrane, room acoustics, electrospinning

## Abstract

Solitary sound absorbing elements exist; however, their construction is massive and heavy, which largely limits their use. These elements are generally made of porous materials that serve to reduce the amplitude of the reflected sound waves. Materials based on the resonance principle (oscillating membranes, plates, and Helmholtz’s resonators) can also be used for sound absorption. A limitation of these elements is the absorption of a very narrow sound band to which these elements are “tuned”. For other frequencies, the absorption is very low. The aim of the solution is to achieve a high sound absorption efficiency at a very low weight. A nanofibrous membrane was used to create high sound absorption in synergy with special grids working as a cavity resonator. Prototypes of the nanofibrous resonant membrane on a grid with a thickness of 2 mm and an air gap of 50 mm already showed a high level of sound absorption (0.6–0.8) at a frequency of 300 Hz, which is a very unique result. Since acoustic elements, i.e., lighting, tiles, and ceilings, are designed for interiors, an essential part of the research is also the achievement of the lighting function and the emphasis on aesthetic design.

## 1. Introduction

Acoustic elements, i.e., tiles, and ceilings, are generally made of porous materials that serve to reduce the amplitude of the reflected sound wave. Sound absorption means the irreversible transformation of sound energy into another energy. The largest amount of energy is consumed by the friction of oscillating air particles against the pore walls of acoustic elements, where the velocity gradient is reduced in the interlayer, and the kinetic energy of the particles is irreversibly converted into thermal energy. It is known [1] that in the case of fibrous acoustic materials, the diameter of the fibers is a key parameter, because the specific acoustic resistance is inversely proportional to the fourth power of the diameter of the fibers. Thereby, if the diameter of the fibers is reduced by two, the specific acoustic resistance increases sixteen times, which means that the extremely small diameter fibers achieve an extremely high specific acoustic resistance and therefore an unusually high sound dampening ability. In an experimental study [2], the parameters of acoustic wave propagation, i.e., acoustic velocity and acoustic pressure, were derived. The static resistivity was determined based on the measurement of the permeability of bulky fiber materials. The permeability of the materials is again a function of the diameter of the fibers at a constant filling.

Materials based on the resonance principle may also be used for sound absorption, which can be divided into three groups, i.e., arrangements behaving like oscillating membranes, arrangements behaving like oscillating plates, and arrangements based on the principle of Helmholtz’s resonators. A significant limitation of these elements lies in the absorption of a very narrow sound band to which these elements are “tuned”. For other frequencies, the absorption is very low. The influence of the surface mass of the membrane on the resonant frequency of the system, as well as the sound absorption coefficient, which increases with this characteristic, has been experimentally confirmed [3]. These materials are applied as boards either directly to the wall/ceiling or, mostly, at some distance from the wall/ceiling, and they can also be filled. Every newly built or renovated room must address their acoustics for their successful approval and especially future operation. Currently, the acoustic tiles available on the market are mainly made of plasterboard elements and elements based on mineral wool, which make up a very significant percentage of the market. Another group are materials is made from organic fibers, either in the form of primary, higher quality, or secondary raw materials, which in turn are cheaper. Other interior elements for absorbing sound energy are made from material based on wood and wood fiber boards, perforated sheet metal, polyurethane, epoxy-bonded sand, etc. Acoustic parameters are mostly represented by the practical sound absorption coefficient α_p_ (–). However, in the case of solitary objects, it is often necessary to quantify the acoustic properties using the parameter of the equivalent absorbing area A (m^2^) corresponding to a single object.

Nanofibrous layers, produced by electrospinning using a DC voltage source, are spun from a polymer solution or melted onto a support layer, mainly a textile. The production of nanofibrous layers by electrospinning is described in patents [4,5], while [5] is a patent of the Technical University in Liberec, which caused a breakthrough in the production of nanofibers and their mass introduction into industrial production. The unique acoustic properties of nanofibrous layers are given both by the large specific surface of the nanofibers, on which viscous losses of acoustic energy can occur, and also by the ability of the nanofibrous layer to resonate at its own frequency, resulting in the amplitude of the membrane vibrated by sound and further losses of acoustic energy, especially at lower sound frequencies.

Review [6] presents research on micro/nanofiber materials fabricated by electrospinning for sound absorption. Nanofibers have sound absorption benefits due to their small fiber diameter, high specific surface area, and high porosity. A composite made of nanofibers and other porous materials effectively absorbs sound at medium and high frequencies. In addition, nanofibers, together with 3D or Helmholtz resonance structures, can also be considered as sound-absorbing materials in their own right.

The properties of the nanofibrous structure cannot be derived by extrapolation of the microstructure parameters with respect to the dimensions of the interfibrous spaces, which are the boundary between viscous flow and molecular gas flow, where the transition region is defined by the Knudsen number *K* (-):(1)$$K=\frac{\lambda }{d_{p}},$$
where *λ* (m) is the mean free path of molecules between two consecutive collisions, and *d_p_* (m) is the characteristic dimension of the system through which the air flows (in our considered case, it is the diameter of the pores). The behavior of the flow along the nanofibers, where the mean free path of the molecules is comparable to the characteristic dimension, is then determined by the synergy of both mechanisms. According to [7], viscous flow is given by the ratio *λ*/*d_p_* < 0.1, which results in the characteristic dimension of the porous structure *d_p_* > 10λ, i.e., *d_p_* > 670 nm. Conversely, molecular flow occurs in the case of *λ*/*d_p_* > 1, i.e., the characteristic dimension of the porous structure is then *d_p_* < *λ*, i.e., *d_p_* < 67 nm. The nanofibrous structure with its characteristic inter-fibrous dimensions of 100–600 nm corresponds precisely to the transition region between viscous and molecular flow, namely, 67 nm < *d_p_* < 670 nm. The above facts apply to air molecules, the size of which is stated to be 0.37 nm, and the mean free path of the molecules *λ* is then 67 nm. A study [8] stated that when the pore size becomes comparable to the molecular mean free path of a saturating fluid, the non-slip conditions on the pore surface are no longer accurate and hence the slip effects have to be taken into account. Therefore, the sound propagation in microfibrous materials is modeled analytically, approximating the geometry by a regular array of rigid parallel cylinders. It has been shown that the velocity and thermal slip on a cylinder surface significantly change the model predictions, leading to a lower attenuation coefficient and higher sound speed values.

The effects of porosity, thickness, average pore size, compression ratio, and combinations of these parameters on the dynamic flow resistivity and sound absorption coefficient of compressed fibrous porous materials are discussed theoretically in [9]. The predicted sound absorption coefficient for each Biot parameter can be employed in the Limp frame model. The predicted values of the sound absorption coefficient when the flow resistivity, porosity, viscous characteristic length, thermal characteristic length, and bulk density varied were shown and discussed in [10]. If we consider a porous material, we can use Poiseuille’s law to describe the air flow resistivity of a porous material *σ* (Pa.s.m^−2^) as follows [11]:(2)$$\sigma =\frac{k}{h}\frac{8\eta }{r_{p}^{2}},$$
where *k* (-) is the structure factor, *h* (-) is its structure porosity given by the ratio of the air volume to the total volume of the material, *η* (Pa.s) is the viscosity of the flowing medium, and *r_p_* (m) denotes the average radius of the pores of the material. According to [12], the pore radius is then related to the equivalent average fiber radius *r* (m) as follows:(3)$$\frac{r_{p}}{r}=\frac{\delta }{1+q}\frac{h}{1-h},$$
where *δ* (-) is the shape constant of the pore, and *q* (-) is the shape constant of the fiber (for a circular cross-section *q* = 0). It is clear from the given relations (1)–(3) that the size of the fibers, or of the pores, will fundamentally affect the process of sound absorption by submicron structures.

If a sound wave hits the nanofibrous membrane, it will cause it to undergo forced oscillations, the amplitude of which is maximal in the case of resonance, as described in [13]. The analysis of sound energy ratios for three boundary structures shows that the film oscillates when a perpendicular longitudinal sound wave hits it and a large amount of energy is reflected back from it; part of it is emitted behind material, and a narrow band around the resonance frequency is absorbed. Most of the energy passes through the thin microstructure; only a small part is absorbed in the structure. The nanofibrous structure oscillates, absorbing both the vicinity of the resonance frequency and higher frequencies in its structure, as depicted in Figure 1. These assumptions correspond to the results in Figure 2 and Figure 3, which show the different absorption curves of the film, the microporous structure, and the nanofibrous structure. These thin materials do not have the same areal weight because it is technologically impossible to produce these different structures with a coherent area. The foil absorbs in a narrow frequency band in accordance with the theoretical assumption that the micro-structure (carrier for the nanofibrous membrane) absorbs high frequencies above 1000 Hz and the nanofibrous structure absorbs both lower frequencies around 500 Hz, where the foil is effective, and higher frequencies where the microporous structure is effective, thereby creating a synergistic effect between the two mechanisms, which other structures do not achieve.

The design of the new acoustic system described in this study was based on the input values of the sound frequency spectrum and the geometry of the space being addressed. A music hall, a large open-space office, or a sports field all show different frequency spectra. From these inputs, the controlled values of the parameters of the acoustic element must be used to achieve the required values of the output acoustic characteristics for the specific space being addressed. The further development of these materials is currently aimed at applications in spatial acoustics, where the carrying capacity of acoustic systems is limited, as well as applications with an emphasis on lighting aesthetics and design. The output of the solution is an acoustic body with a lighting function for applications in spatial acoustics. The nanofibrous layer is applied to a suitable carrier material, preferably provided with perforations of various shapes and sizes, which allows it to resonate with the resulting sound absorption in a wide frequency spectrum. This surface structure, which must be translucent and also sufficiently flexible, is further spatially shaped.

The nanofibrous layer is applied to the surface of a flexible structure preferably equipped with a specific perforation. This flat structure is rolled into a three-dimensional body with the final shape of a cylinder with axial or peripheral lighting. The solution consists of the design and optimization of structural parameters, in particular the area weight of the nanofibrous layer; the thickness of the spatial structure indicating the distance between individual nanofibrous resonators; the filling, size, and shape of the pores of the spatial structure; and the number of windings indicating the total effective area of the nanofibrous membrane. Part of the solution is also the determination of optimal adhesive agents. The nanofibrous membrane in this element is prepared by the method of electrostatic spinning of a Nanospider™ polymer solution using a direct voltage source. Alternatively, part of the interior space of the structure may be filled with a bulky nanofibrous structure using the electrostatic spinning method using an AC voltage source without grounding, or the centrifugal spinning method. The distance of the individual layers during the winding of the flexible structure into the resulting spatial arrangement is achieved in at least one case using a light LED cable with a diameter of 2–5 mm. The significant translucency of the developed acoustic element with a nanofiber structure makes it possible to create a very pleasant diffused light source, i.e., a luminous acoustic body.

## 2. Materials and Methods

### 2.1. Nanofibrous Structure

Three methods for nanofibrous structure spinning were used for acoustic sample preparation. For the production of PA6 nanofibrous membranes, the cord electrospinning method was used [15]. In this method, the cord was connected to a high voltage supply, and at the top of the cord there was a counter electrode, which was grounded. A polymeric solution was applied onto the cord around its whole circumference, and then the application means moving reversibly along the active spinning zone of the cord, and the process of electrostatic spinning of the liquid polymeric material were initiated. Taylor cones were created on the cord surface towards the counter electrode. The nanofibers produced on the NS 8S1600U line covered the spun bond carrier with a basis weight of 30 g/m^2^ during the electrospinning process. The average fiber diameter was 162 ± 48 nm, and the base weight of the nanofibrous membrane was 0.22 ± 0.03 g/m^2^ (PA6022). The next nanofibrous layer was spun using PARDAM NANO4FIBERS (Roudnice nad Labem, Czech Repubpic) centrifugal spinning from a PVDF melt at 130 °C with the resulting basis weights of 5, 10, 16, and 45 g/m^2^ (PVDF-5/10/16/45) onto a PP carrier with a basis weight of 50 g/m^2^ according to the patent [16]. The average fiber diameter was 350 ± 150 nm. Inorganic SiO_2_ fibers with a total weight of 17 g were produced using PARDAM NANO4FIBERS centrifugal spinning according to the patent [17]. The free spaces in the M101 grid were filled with a SiO_2_ nanofibrous structure crushed to about 2 mm. The average SiO_2_ nanofiber diameter was 200 ± 100 nm. Figure 4 shows the nanofibrous structure spun directly onto one of the grids and on a ring.

### 2.2. Grid Construction

In the AutoDesk Fusion 360 program, complex patterns were modified for grids, which were subsequently printed on an Original PRUSA I3 MK3S 3D printer (Prague, Czech Republic) in the form of wheels with a frame diameter of 106 mm or 32 mm so that they could then be inserted together with the nanofiber membrane onto the front of a two-microphone impedance tube for determining the sound absorption coefficient in the frequency band 100–1600 Hz (100 mm inner diameter of the frame) or 500–6400 Hz (29 mm inner diameter of the frame). The inner diameter of the grid frame corresponded to the inner diameter of the tube, and consequently the diameter of the active surface for determining the sound absorption coefficient. For the development, 110 grid constructions were tested with different types of nanofibrous membranes. Ten 2 mm thick and one 10 mm thick grid constructions were chosen to demonstrate the acoustic data (see the images in Figure 5). The 10 mm thick sample labelled grid M101 was filled with a bulky nanofibrous structure.

The porosity was calculated as 1-filling, which was determined by the AutoDesk Fusion 360 program as the ratio of the content of the solid part to the content of the whole unit. The porosity of the grid itself (100 mm in diameter) was 0.662 for M21, 0.476 for M7, 0.674 for M27, 0.606 for M13, and 0.807 for M101, and the ring R had a full active area. The porosity of the grid itself (100/29 mm of diameter) was 0.688/0.678 for M1, 0.684/0.702 for M2, 0.669/0.622 for M3, and 0.667/0.656 for M4, respectively. The non-flammable PVAc dispersion adhesive Collano DW 2040 (Sempach Station, Switzerland) was used for the permanent connection of the grid and the nanofibrous membrane spun onto a carrier. In the case of the volume nanofibrous structure, glue was first applied to the selected grid (M101) on one side, and then this adhesive layer was pressed onto the nanofiber layer applied to the carrier so that the stacking was in the following order: carrier–PA6022–glue–grid. The openings of the grid were filled with bulky SiO_2_ nanofibers, and the free side was closed again with a nanofiber layer applied to the carrier so that the overall assembly was in the following order: carrierSB30–PA6022–glue–grid–glue–PA6022–carrierSB30 (with cavities filled with a bulky nanofibrous structure of SiO_2_). In one experiment, the grids were provided with a nanofibrous membrane on both sides.

### 2.3. Sound Absorption Measurement

A two-microphone impedance tube type 4206 (Brüel & Kjær, Nærum, Denmark) was used to determine the sound absorption coefficient according to the ISO 10534–2 standard. A sound source (loudspeaker) was mounted at one end of the impedance tube, and a sample of the material was placed at the other end, as shown in Figure 6. The loudspeaker generated broadband, stationary random sound waves, which propagated as plane waves in the tube hit the sample and reflected. The propagation, contact, and reflection resulted in a standing-wave interference pattern due to the superposition of forward- and backward-travelling waves inside the tube. By measuring the sound pressure at two fixed locations and calculating the complex transfer function using a two-channel digital frequency analyzer, it was possible to determine the sound absorption and complex reflection coefficients and the normal acoustic impedance of the material. The usable frequency range depended on the diameter of the tube and the spacing between the microphone positions. The sound absorption coefficient of materials at a perpendicular incidence of sound was determined. The front side of the tested material was located in the so-called reference plane.

The sound absorption coefficient α is defined by the ratio of the absorbed acoustic energy to the total incident energy. Its size is given by an interval from 0 to 1, when for a perfectly absorbing body it acquires α = 1 (100% absorptivity) and for a perfectly reflective body α = 0 (no absorptivity). Good absorbing materials are those for which α is higher than 0.6.

The method measures one sample with two different surfaces, or diameters, namely, 100 mm for determining the sound absorption coefficient from 100 Hz to 1600 Hz, and 29 mm from 500 Hz to 6400 Hz. The result is usually a combined curve. For resonant elements, a combination of both measurements may not be appropriate, as the resonant behavior depends on the shape and size of the sample. This is why both curves are shown separately in this study. A sample with a thickness of 1–2 mm was placed in a reference plane with an air gap so that the face of each sample (side of sound impact) was always in the same position or distance from the microphones in the so-called reference plane. The sample holder (piston) was therefore always moved to a distance of 50 mm from the reference plane so that the required air gap was created between the sample and the reflective surface of the piston.

The special grids produced on a 3D printer were installed together with the nanofibrous membrane on a thin carrier textile in an impedance tube and measured with an air gap of 50 mm in the following order in the direction of sound incidence: grid–adhesive–nanofibrous structure–spun bond carrier.

## 3. Results

This section describes, among other things, the dependence of sound absorption α (–) on sound frequency f (Hz). The results are divided into subsections according to the sample parameters that were varied. These were the pattern, basis weight, number of nanofiber resonators, and volume filling.

### 3.1. Pattern Influence on Sound Absorption

Figure 7 and Figure 8 show that the nanofibrous structure applied to the grid structure (colored curves) had a significantly high absorption in the entire measured frequency spectrum compared to the nanofibrous structure without a grid (only in the frame = black curve). The nanofibrous structure from PA6 on a spun bond carrier had a weight of 0.22 ± 0.03 g/m^2^ (PA6022).

Figure 7 compares the sound absorption coefficient of four randomly selected grids with a nanofibrous membrane. For clarity of results, only sound frequencies measured from samples with a large diameter are shown, i.e., 100 mm sample diameter for 100–1600 Hz.

As shown in Figure 7, the influence of the different grid patterns with a constant nanofibrous membrane on the sound absorption factor was significantly large. The samples showed very different acoustic behaviors. This was due to the synergy effect of the grid with the nanofibrous membrane, where the size and shape of the individual surfaces that bound the local resonators defined the behavior of these membrane resonators. Figure 8 shows the best acoustic systems where the samples of the nanofibrous structure on a grid with a thickness of 2 mm and an air gap of 50 mm already showed high sound absorption (0.6–0.8) at a frequency of 300 Hz, which is a very unique result. From the comparison of the curves of the measured samples with two different diameters for the frequency range 100–6400 Hz, it is clear that the filling (ratio of the grid area to the total sample area) of both types was different and therefore could not be compared. This did not apply to the nanofibrous membrane clamped in a ring of both diameters, where in both cases the entire area of the nanofibrous membrane was used, and the sound absorption coefficient curves from both measurements were perfectly related to each other (see black curves in Figure 8). The acoustic systems where a grid divided the nanofibrous membrane into individual locations in which the nanofiber membrane oscillated with the boundary conditions given by the size and shape of the grid mesh were already effective at 300 Hz (see colored curves in Figure 8) compared to the nanofiber layer itself installed only in the frame/ring (see black curve in Figure 8).

### 3.2. Basis Weight Influence on Sound Absorption

Figure 9 shows that the nanofibrous structures from PVDF with basis weights of 5, 10, 16, and 45 g/m^2^ (PVDF-5/10/16/45 from centrifugal spinning) had significant effects on sound absorption. For completeness, the results were compared with a PA6 nanofibrous membrane with an area weight of 0.22 g/m^2^ (PA6022 from electrospinning).

Figure 9 also shows the significant effect of the basis weight of the nanofibrous layers on the sound absorption coefficient. Samples of grid M2 with a nanofibrous layer made by electrospinning from PA6 (PA6_0.22 g/m^2^) showed broad-spectrum acoustic efficiency, i.e., unique acoustic properties with a minimum sample thickness (2 mm) and an air gap of only 50 mm. They achieved high sound absorption from approximately 350 Hz, which is again a very exceptional result with such a low sample thickness and the entire acoustic system. Similar results were achieved by the grid M2 samples with a nanofibrous layer made using a centrifuge from PVDF (PVDF 5/10/16 g/m^2^). The highest basis weight (45 g/m^2^) centrifuge-produced PVDF samples showed only narrowband absorption and therefore insufficient acoustic results. This nanofibrous layer ceased to have an open structure and behaved similarly to a homogeneous film. In general, the first local maximum of the sound absorption factor shifts towards lower sound frequencies with increasing surface mass of the nanofibrous resonant membrane, or the natural frequency of the acoustic system is lower as the basis weight of the nanofibrous resonant membrane increases. This prediction was not completely valid for our samples, because there was a synergy between the cavity resonator (grid, perforated plate, etc.) and the nanofibrous membrane resonator, and the resulting acoustic properties of the entire acoustic system could not be predicted. It is interesting that samples with a very low surface weight (black_0.25 g/m^2^) produced by electrospinning and samples with a much larger surface weight produced by centrifugal spinning (yellow_10 and green_16 g/m^2^) showed similar acoustic behaviors. This was mainly due to the diameter of the nanofibers and different structures of nanofibrous layers. Figure 10 shows the same effect of the areal weight of the nanofibrous layer on the sound absorption, with the nanofibrous samples installed in the R ring. However, in this case, the low basis weight electrospinning nanofibrous membrane itself (PA6022) did not show significant sound absorption. In order for a nanofibrous membrane with a low basis weight to be effective, its resonant behavior must be supported by defining the individual resonant surfaces using a grid. The basis weight of the nanofibrous layers produced by centrifugal spinning (PVDF 5/10/16/45) was sufficient for absorption of medium frequencies (1000–2000 Hz), but the samples did not show absorption of lower frequencies (approximately 300–1000 Hz), where again there was no support for the resonant behavior of the local areas delimited by the grid.

From a comparison of Figure 9 and Figure 10, the synergy effect of the nanofibrous membrane with the grid is obvious. These acoustic systems were already effective at 300 Hz compared to the nanofiber layer itself installed only in the frame without grid divisions into individual locations, in which the nanofiber membrane oscillated with the boundary conditions given by the size and shape of the grid mesh.

### 3.3. Grids with Nanofibrous Structures on Both Sides

A nanofibrous structure including the carrier was laminated on five grid constructions (see Figure 5) provided with Collano glue. The nanofiber layer was spun from a PVDF solution with a resulting basis weight of 5 g/m^2^ (PVDF5) onto the carrier. Similarly, a nanofibrous layer of PA6 with basis weight of 0.25 g/m^2^ (PA6025) was produced for application to the other side of the grid structure. The sample was measured from both sides in the following order from the direction of sound incidence:carrierPP50–PVDF5–glue–grid–glue–PA6025–carrierSB30;carrierSB30–PA6025–glue–grid–glue–PVDF5–carrierPP50.

Figure 11 shows that there was no significant difference in the order of the different nanofibrous structure applied to both sides of the grid structure (green vs. red curve). From a comparison of samples with nanofibrous structures on both sides (green + red curve) and samples with only one-sided application of the same nanofibrous structures (black + blue curve), an increase in absorption at lower frequencies (around approximately 350 Hz) could be seen, but together with a decrease in sound absorption at higher frequencies (from approximately 500 Hz). The benefit of a double-sided application is therefore not significant and from a cost point of view, the implementation of this type of acoustic system does not make sense.

### 3.4. Grid Filled by the Volume Nanofibrous Structure

In the case of grid M101 (see Figure 12), the free space was filled by the volume nanofibrous structure, and both faces were covered by the nanofibrous structure from PA6 on a spun bond carrier (PA6022). Subsequently, the sample was measured with different air gaps between the sample and the piston inside an impedance tube.

## 4. Discussion and Conclusions

The synergy effect of the nanofibrous membrane with the grid was confirmed. The acoustic systems where a grid divides the nanofibrous membrane into individual locations, in which the nanofiber membrane oscillates, with the boundary conditions given by the size and shape of the grid mesh are already effective at 300 Hz compared to the nanofiber layer itself installed only in the frame/ring. It was shown how the boundary conditions given by the size and shape of the grid are important for the resonant behavior of the nanofiber membrane.

The samples composed of both a ring and grid were provided with a nanofibrous layer of different basis weights (PA6 0.25 g/m^2^/PVDF 5/10/16/45 g/m^2^). Grid samples with an applied nanofiber layer show a significant effect of the area weight of the nanofiber layers on the sound absorption coefficient. They achieve high acoustic efficiency from approximately 300 Hz. There is a synergy between the cavity resonator (grids, perforated plates, etc.) and the nanofibrous membrane resonator. The resulting acoustic properties of the entire acoustic system cannot be predicted. We found that there is no significant difference in the order of the different nanofibrous structures applied to both sides of the grid structure. From a comparison of samples equipped with nanofibrous structures on both sides and samples with only one-sided application of the same nanofibrous structures, an increase in sound absorption at lower frequencies (around approximately 350 Hz) is visible, but together with a decrease in sound absorption of higher frequencies (from approximately 500 Hz). The benefit of double-sided application is therefore not significant. The construction, filled with a bulky structure and closed with a nanofiber membrane on both sides, does not show sufficient sound absorption.

For the future development of spatial acoustic bodies, there is a sufficient amount of results and knowledge about nanofibrous structures applied to complex grid structures. The solution is aimed at applications in room acoustics, where the carrying capacity of acoustic systems is limited. It also includes applications with an emphasis on lighting aesthetics and design. A nanofibrous layer will be applied to a flexible grid structure that will create an aesthetic pattern when the lighting is turned on. This assembly is equipped with a light source and rolled into the resulting shape of a spatial acoustic body. The distance between the individual layers after rolling is ensured either only by a light cable, or by spacers, or by a translucent flat material of the required thickness.

## Figures and Tables

**Figure 1 nanomaterials-13-01115-f001:**
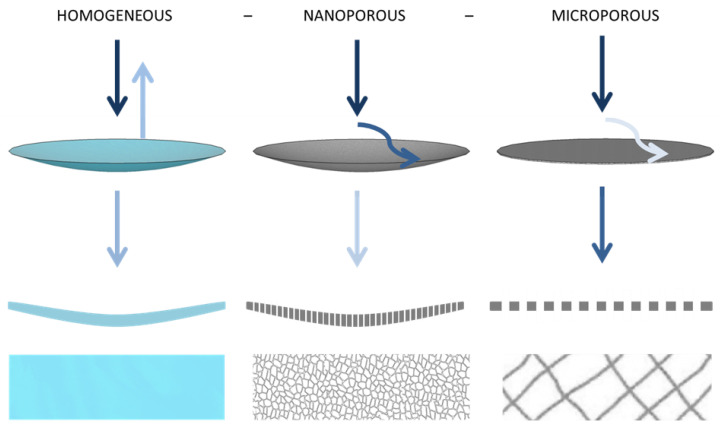
Diagram of the principle of sound absorption by the thin layers of a homogeneous film, a nanoporous membrane, and a microporous layer.

**Figure 2 nanomaterials-13-01115-f002:**
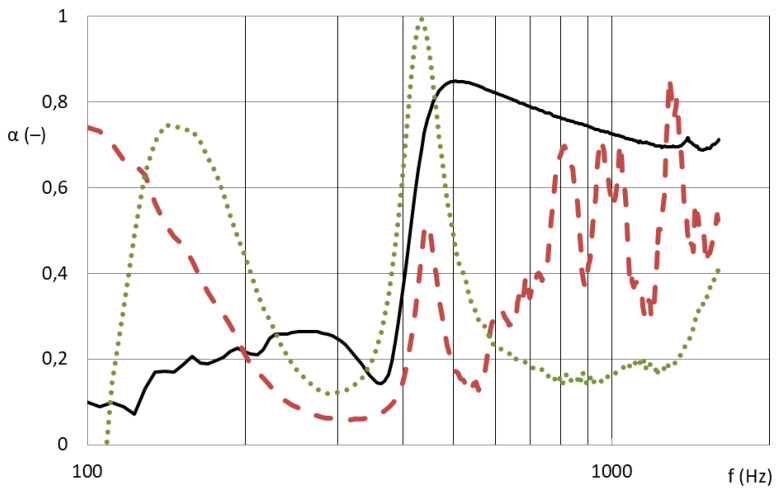
Frequency dependence of the sound absorption coefficient; quad-perforated plate of 9 mm size/11 mm span is covered by a nanofibrous membrane of 0.2 g/m^2^ on a carrier of 25 g/m^2^ (black line), or foil of 7 g/m^2^ (red dashed line), or foil of 40 g/m^2^ (green dotted line). The air gap between the 1 mm thick panel and the wall is 50 mm [13].

**Figure 3 nanomaterials-13-01115-f003:**
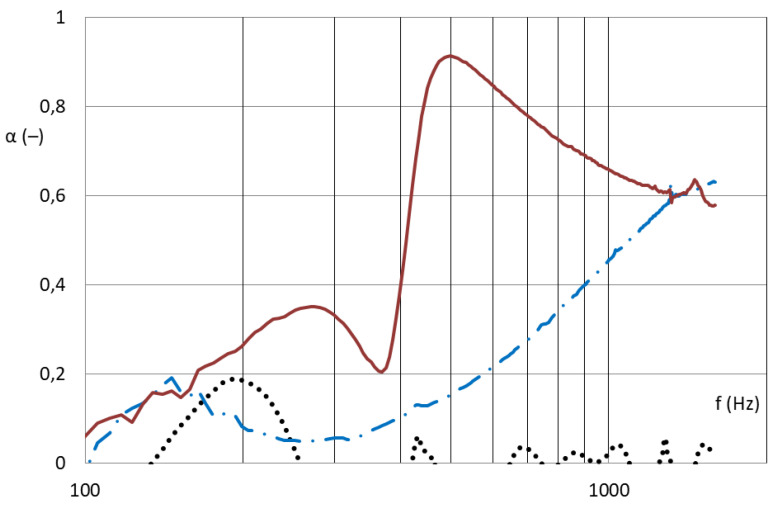
Frequency dependence of the sound absorption coefficient; aluminum perforated sheet with a thickness of 1 mm, with square centered holes with a hole diameter of 8 mm, hole spacing of 10 mm (black dotted line), the same perforated sheet with a nanolayer carrier (blue dashed line) and a perforated sheet with a nanolayer on a carrier (red line). All the samples were placed at a distance of 50 mm from wall [14].

**Figure 4 nanomaterials-13-01115-f004:**
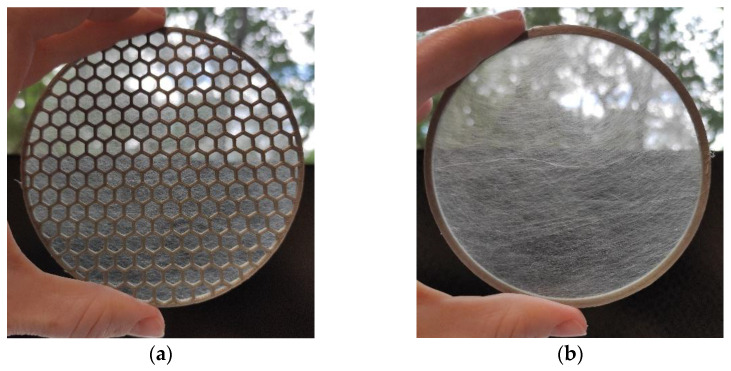
Images of a PA6 nanofibrous structure on: (**a**) grid M2; (**b**) grid R (ring).

**Figure 5 nanomaterials-13-01115-f005:**
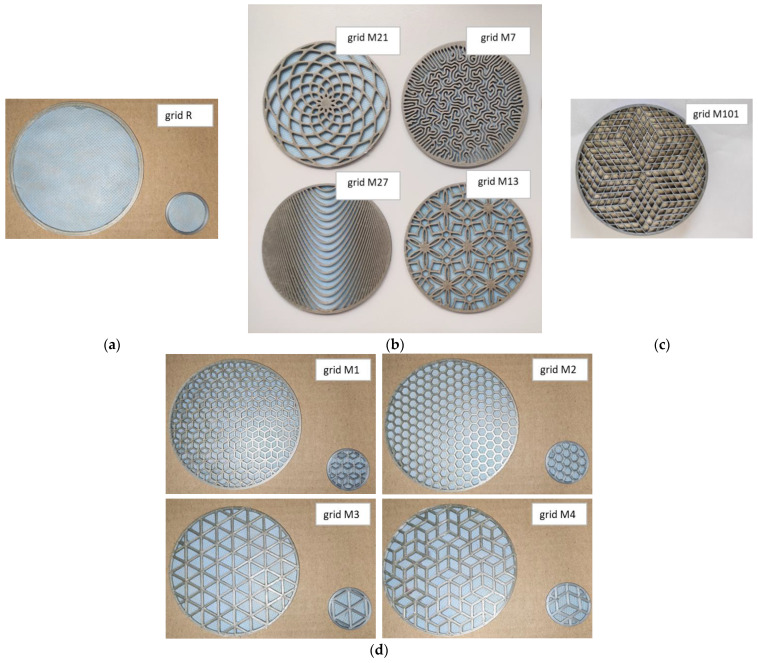
Images of selected samples of grid: (**a**) nanofibrous membrane in the ring R; (**b**) randomly selected patterns M21, M7, M27, M13; (**c**) grid filled with a bulky nanofibrous structure M101; (**d**) patterns with the best sound absorption coefficient M1, M2, M3, M4.

**Figure 6 nanomaterials-13-01115-f006:**
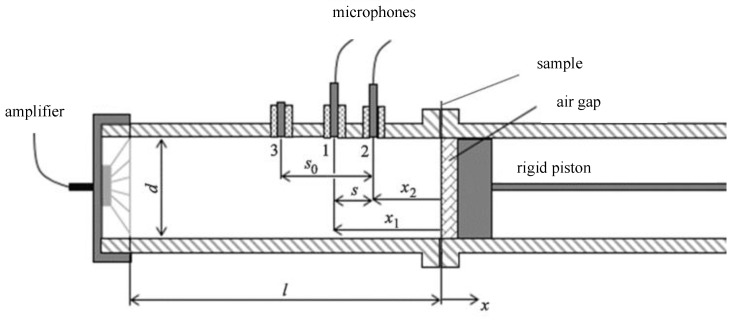
Diagram of measurement inside the two-microphone impedance tube type 4206.

**Figure 7 nanomaterials-13-01115-f007:**
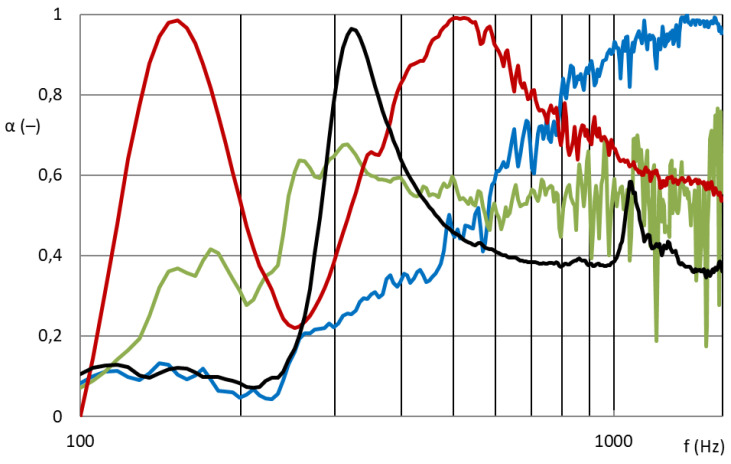
Sound absorption coefficient depending on sound frequency (100–1600 Hz). Nanofibrous membrane **PA6022** on the grid; blue: M21; green: M7; red: M27; black: M13.

**Figure 8 nanomaterials-13-01115-f008:**
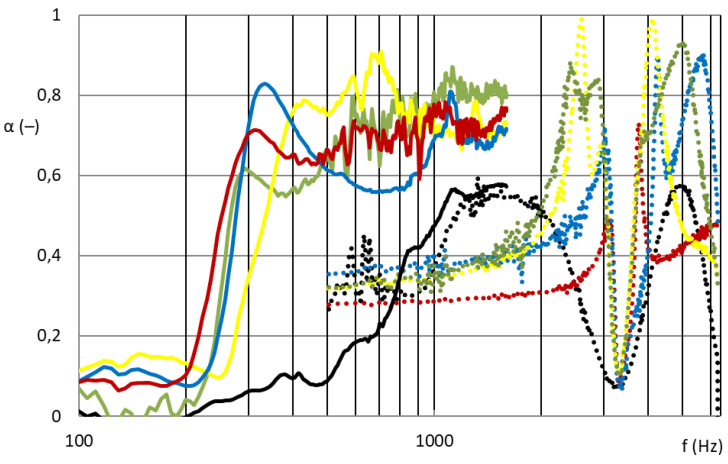
Sound absorption coefficient depending on sound frequency (100–6400 Hz). Solid curve: 100 mm diameter samples for frequency range 100–1600 Hz; dotted curve: 29 mm diameter samples for frequency range 500–6400 Hz. Nanofibrous membrane **PA6022** on the grid; red: M1; blue: M2; yellow: M3; green: M4; black: R.

**Figure 9 nanomaterials-13-01115-f009:**
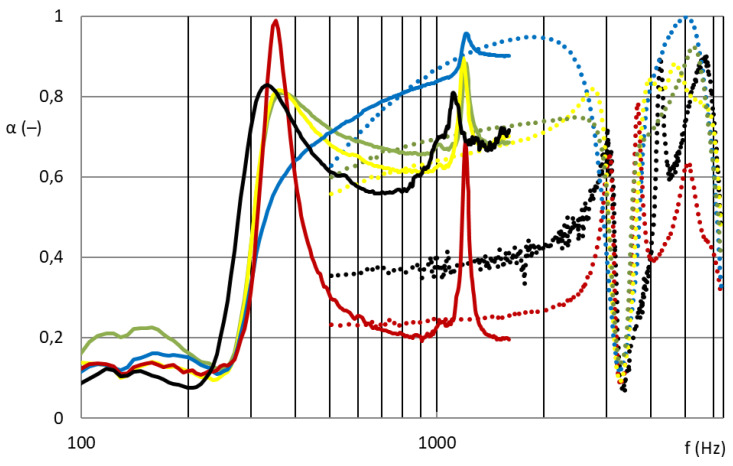
Sound absorption coefficient depending on sound frequency (100–6400 Hz). Solid curve: 100 mm diameter samples for frequency range 100–1600 Hz; dotted curve: 29 mm diameter samples for frequency range 500–6400 Hz. **Grid M2** with a nanofibrous membrane: blue: PVDF5; yellow: PVDF10; green: PVDF16; red: PVDF45; black: PA6022.

**Figure 10 nanomaterials-13-01115-f010:**
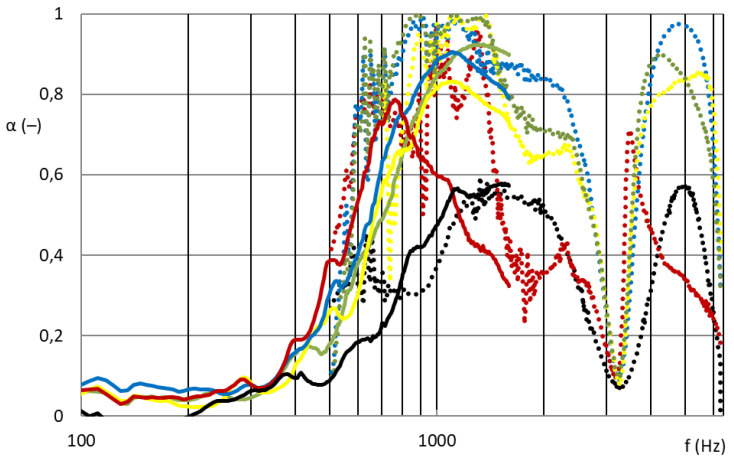
Sound absorption coefficient depending on sound frequency (100–6400 Hz). Solid curve: 100 mm diameter samples for frequency range 100–1600 Hz; dotted curve: 29 mm diameter samples for frequency range 500–6400 Hz. **Ring R** with nanofibrous membrane: blue: PVDF5; yellow: PVDF10; green: PVDF16; red: PVDF45; black: PA6022.

**Figure 11 nanomaterials-13-01115-f011:**
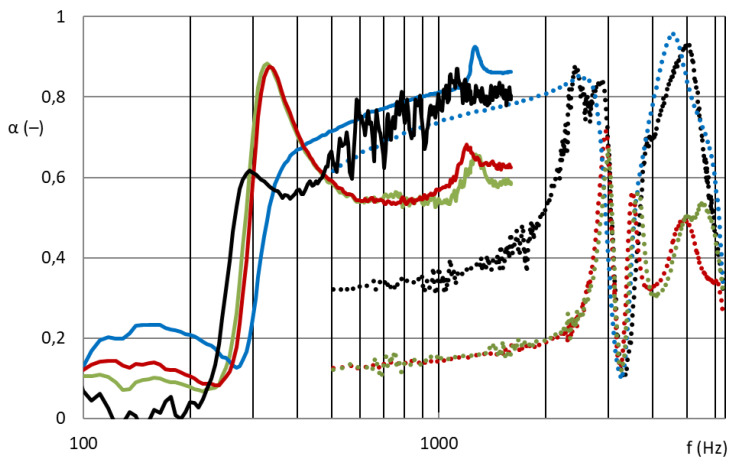
Sound absorption coefficient depending on sound frequency (100–6400 Hz). Solid curve: 100 mm diameter samples for frequency range 100–1600 Hz; dotted curve: 29 mm diameter samples for frequency range 500–6400 Hz. **Grid M4** with nanofibrous membrane: blue: PVDF5; black: PA6022; green: PA6022 and PVDF5; red: PVDF5 and PA6022.

**Figure 12 nanomaterials-13-01115-f012:**
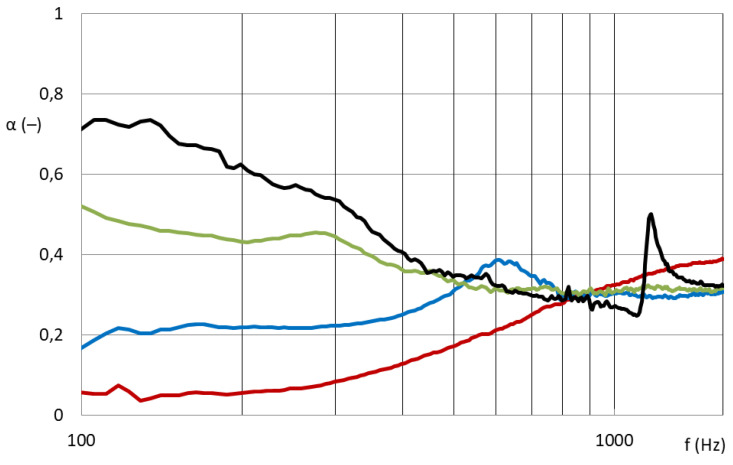
Sound absorption coefficient depending on sound frequency (100–1600 Hz). **Grid M101** with a nanofibrous membrane from both sides and filled with the volume nanofibrous structure measured with different air gaps: red: 0 mm; blue: 50 mm; green: 100 mm; black: 150 mm.

## Data Availability

Data sharing is not applicable to this article.

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
