# Peer review of "The Application of Nanofibrous Resonant Membranes for Room Acoustics"

_nanomaterials, 2023, doi:10.3390/nano13061115_

Round 1
Reviewer 1 Report
The paper deals with the sound absorption of resonant membranes incorporating nanofibers, presenting an experimental study on the effect of such membranes used together with a rigid grid, with different patterns. The paper is interesting and presents novel information. I recommend publication following addressing the following issues:
- There is an extensive literature on the behaviour of different absorbing systems, including metamaterials and innovative solutions in line with the ones presented. Perhaps a more up-to-date literature review should be made, specially taking into account that from the papers listed only 3 are from the last 5 years.
- In the impedance tube scheme, please also represent the sample and the dimensions of the air cavity tested, so that is is more clear for readers.
- In Fig 7, perhaps a more detailled comment on the relation between the grid dimensions and the resonance frequencies seen in the plot should be given.
- Please comment why the results for the two different diameters of the samples are so different, for the same frequencies, such as in Figures 8 and 11.
Author Response
Dear Reviewer,
Thank you for your comments.I have edited the manuscript as follows:
- There is an extensive literature on the behaviour of different absorbing systems, including metamaterials and innovative solutions in line with the ones presented. Perhaps a more up-to-date literature review should be made, specially taking into account that from the papers listed only 3 are from the last 5 years. References 6 and10 have been added.
- In the impedance tube scheme, please also represent the sample and the dimensions of the air cavity tested, so that is is more clear for readers. It was modified.
- In Fig 7, perhaps a more detailled comment on the relation between the grid dimensions and the resonance frequencies seen in the plot should be given. Unfortunately, prediction of resonant frequencies for such complex patterns is not possible because the nanofibrous membrane resonates within a single complex pattern apparently over a wide frequency spectrum.
- Please comment why the results for the two different diameters of the samples are so different, for the same frequencies, such as in Figures 8 and 11. The filling (ratio of the grid area to the total sample area) of both types is different and therefore cannot be compared.
Reviewer 2 Report
Review of The Application of Nanofibrous Resonant Membranes for Room Acoustics
The manuscript presents research on how to implement resonant membranes through nanofibrous techniques for room acoustics applications. The topic is interesting as it provides a nanofibrous technique implemented with rigid customised grids to investigate the related implications. The introduction provides an extensive yet not selected overview of the theme and does not sufficiently highlight it. The methodology is described well, and the results are presented in an extensive, again, not yet selected way, with the presentation of figures and tables to support/illustrate the findings; however, they could be expressed more clearer. Moreover, I would recommend a revision of the English. I also have some specific comments:
1. Abstract: please avoid terms regarding novelty or significance such as unique, new, original, novel, important, and significant.
2. Abstract: I think that the main overall aim could be specified better. Indeed, I think the research results are interesting, so lines 19-21 are a bit reductive and could be extended to more specific applications for both research and design.
3. English should be improved, and I think proofreading would benefit the overall readership of the paper.
4. Introduction: I think a lack of references undermines the paper's robustness. Please, include more relevant references from the field of acoustics, nanofibers and any other relevant ones.
5. For the sake of the readership, references from the Nanomaterials MDPI journal should be added to prove the reliability of the paper's discussed topic within the journal.
6. Row 26, specify which sort of acoustic elements.
7. Row 37,38, please specify better which parameters and how they are relevant to your study.
8. Row 47,48, Please avoid adjectives such as obvious and stick to the scientific, academic English tone.
9. Overall I think that the Introduction is too broad and dispersive. Please try to focus on describing the background fields, whether materials or methods which support your point of view and research question.
10. Introduction: If the figures and equations are relevant for describing the materials and methods, please move them to the appropriate section.
11. While presenting figures and tables, please double-check if they have been presented in other publications, as I think that MDPI policy on Figures and Tables relates strictly to brand-new images or tables. You can refer to them by citing the original source or modifying the original figure, I guess.
12. Materials and Methods, please include relevant references regarding the method and the instruments used for the prototyping and experimental measurements.
13. Rows 211-213, please specify the model of the 3D printer, the main printing setups, and the materials used, adding the corresponding relevant references.
14. Rows 233-236, please specify how you determined each sample's porosity.
15. Sound Absorption measurements, Include references regarding the measurement instruments specifics and the related ISO.
16. Figure 6, could you substitute the technical schematic with a photo taken in the lab of your instrumentation?
17. Rows 285,286, This paragraph is too reductive. I think you should either consider removing it or extending it in relation to what follows next.
18. Rows 287-296. I think your results should be better commented including an observation on why the nanofibrous structure applied to the grid structure (coloured curves) has significantly high absorption in the entire measured frequency spectrum than the nanofibrous structure without a grid. Please do the same with rows 293-296
19. Rows 305-318, this part could be implemented and split between rows 287-296 and 305-318.
20. Rows 305-318, is there a way to quantify the increased sound absorption gap between the different designs in terms of amplitude and frequency? Could you specify it better in this section or include another connected section on this result?
21. Figure 9. I think that the two different-diameter samples are hardly comparable here. Please consider separating the results' analysis or defining a coefficient which could make them comparable in the same graph. Please apply it also to Figures 10 and 11.
22. Results, Overall, I think this section could be presented more clearly, and perhaps, defining a common way of presenting the two different diametres samples and condensing the essential part of the results while avoiding repetitions.
For these reasons, I would recommend a major revision.
Author Response
Dear Reviewer,
Thank you for your comments.I have edited the manuscript as follows:
- Abstract: please avoid terms regarding novelty or significance such as unique, new, original, novel, important, and significant. Some of terms have been removed.
- Abstract: I think that the main overall aim could be specified better. Indeed, I think the research results are interesting, so lines 19-21 are a bit reductive and could be extended to more specific applications for both research and design. Acoustic elemnets have been refined.
- English should be improved, and I think proofreading would benefit the overall readership of the paper. English was corrected.
- Introduction: I think a lack of references undermines the paper's robustness. Please, include more relevant references from the field of acoustics, nanofibers and any other relevant ones. References 6 and10 have been added.
- For the sake of the readership, references from the Nanomaterials MDPI journal should be added to prove the reliability of the paper's discussed topic within the journal. References 6 and10 have been added.
- Row 26, specify which sort of acoustic elements. Acoustic elemnets have been refined.
- Row 37,38, please specify better which parameters and how they are relevant to your study. Acoustic parameters have been listed.
- Row 47,48, Please avoid adjectives such as obvious and stick to the scientific, academic English tone. ok
- Overall I think that the Introduction is too broad and dispersive. Please try to focus on describing the background fields, whether materials or methods which support your point of view and research question. Introduction have been reduced.
- Introduction: If the figures and equations are relevant for describing the materials and methods, please move them to the appropriate section. The figures and equations are included in the introduction chapter to understand the issues.
- While presenting figures and tables, please double-check if they have been presented in other publications, as I think that MDPI policy on Figures and Tables relates strictly to brand-new images or tables. You can refer to them by citing the original source or modifying the original figure, I guess. References are given in square brackets for all cited figures 2 and 3.
- Materials and Methods, please include relevant references regarding the method and the instruments used for the prototyping and experimental measurements. Type of lines has been added.
- Rows 211-213, please specify the model of the 3D printer, the main printing setups, and the materials used, adding the corresponding relevant references. Type of 3D printer has been added.
- Rows 233-236, please specify how you determined each sample's porosity. Porosity calculation has been refined.
- Sound Absorption measurements, Include references regarding the measurement instruments specifics and the related ISO. This specification is listed in the chapter on the first line.
- Figure 6, could you substitute the technical schematic with a photo taken in the lab of your instrumentation? The scheme is, in my opinion, sufficient to describe the method. It was modified.
- Rows 285,286, This paragraph is too reductive. I think you should either consider removing it or extending it in relation to what follows next. This paragraph has been extended.
- Rows 287-296. I think your results should be better commented including an observation on why the nanofibrous structure applied to the grid structure (coloured curves) has significantly high absorption in the entire measured frequency spectrum than the nanofibrous structure without a grid. Please do the same with rows 293-296. The explanation has been added [The acoustic systems where a grid divides the nanofibrous membrane into individual locations, in which the nanofiber membrane oscillates, with the boundary conditions given by the size and shape of the grid mesh are already effective at 300 Hz (see colored curves on the figure 8) compared to the nanofiber layer itself installed only in the frame/ring (see black curve on the figure 8).]
- Rows 305-318, this part could be implemented and split between rows 287-296 and 305-318. The first paragraph of the chapter 3.1. Pattern Influence on Sound Absorption is introduction to theme. The following is a discussion of each chart separately.
- Rows 305-318, is there a way to quantify the increased sound absorption gap between the different designs in terms of amplitude and frequency? Could you specify it better in this section or include another connected section on this result? Unfortunately, prediction of resonant frequencies for such complex patterns is not possible because the nanofibrous membrane resonates within a single complex pattern apparently over a wide frequency spectrum.
- Figure 9. I think that the two different-diameter samples are hardly comparable here. Please consider separating the results' analysis or defining a coefficient which could make them comparable in the same graph. Please apply it also to Figures 10 and 11. The graph with both averages just shows how important the boundary conditions given by the size and shape of the grid are for the resonant behaviour of the nanofibre membrane. In addition, the ratio of the solid phase to the whole formation is different for the two diameters.
- Results, Overall, I think this section could be presented more clearly, and perhaps, defining a common way of presenting the two different diametres samples and condensing the essential part of the results while avoiding repetitions. It was reduced.